# Potential Effects of Regulating Intestinal Flora on Immunotherapy for Liver Cancer

**DOI:** 10.3390/ijms241411387

**Published:** 2023-07-13

**Authors:** Xiangdong Yan, Liuhui Bai, Ping Qi, Jin Lv, Xiaojing Song, Lei Zhang

**Affiliations:** 1The First Clinical Medical College, Lanzhou University, Lanzhou 730000, China; yanxd21@lzu.edu.cn (X.Y.); bailh21@lzu.edu.cn (L.B.); 2Department of General Surgery, The First Hospital of Lanzhou University, Lanzhou 730000, China; qip21@lzu.edu.cn (P.Q.); lvj21@lzu.edu.cn (J.L.); 3Key Laboratory of Biotherapy and Regenerative Medicine of Gansu Province, The First Hospital of Lanzhou University, Lanzhou 730000, China; songxiaojing4227@126.com

**Keywords:** adjuvant therapy of immunity, checkpoint inhibitors for intestinal flora, liver cancer immunotherapy

## Abstract

The intestinal flora plays an important role in the occurrence and development of liver cancer, affecting the efficacy and side effects of conventional antitumor therapy. Recently, immunotherapy for liver cancer has been a palliative treatment for patients with advanced liver cancer lacking surgical indications. Representative drugs include immune checkpoint inhibitors, regulators, tumor vaccines, and cellular immunotherapies. The effects of immunotherapy on liver cancer vary because of the heterogeneity of the tumors. Intestinal flora can affect the efficacy and side effects of immunotherapy for liver cancer by regulating host immunity. Therefore, applying probiotics, prebiotics, antibiotics, and fecal transplantation to interfere with the intestinal flora is expected to become an important means of assisting immunotherapy for liver cancer. This article reviews publications that discuss the relationship between intestinal flora and immunotherapy for liver cancer and further clarifies the potential relationship between intestinal flora and immunotherapy for liver cancer.

## 1. Introduction

Liver cancer is a common malignant tumor worldwide, with the fourth highest incidence and third highest mortality rate in the world [1]. Previously, the treatment of liver cancer was a comprehensive model based on surgical treatment, including radiofrequency ablation, interventional therapy, targeted therapy, chemotherapy, and radiotherapy. However, the postoperative recurrence rate was high, and existing treatment methods had limited effects [2,3,4]. Immunotherapy is a therapeutic method used to control and kill tumor cells by stimulating the immune function and enhancing the antitumor immunity of the tumor microenvironment; the application of immune checkpoint inhibitors (ICIs) has become a current research hotspot [5]. The probability of severe toxic reactions in immunotherapy is lower than in the traditional treatment of liver cancer, and patients are more tolerant to immunotherapy than to traditional treatment. Recently, immunotherapy for liver cancer has shown positive results in clinical settings. After receiving immunotherapy, the survival time of some patients is prolonged, and the recurrence rate is reduced. However, the immunotherapy cycle is lengthy, and patients are often not treated promptly. The clinical efficacy rate of immunotherapy for liver cancer is approximately 20% [6]. Moreover, the intestinal flora and related metabolites, such as bile acids and short-chain fatty acids, can regulate the key immune metabolic processes in chronic hepatitis and liver cancer through the “intestine–liver axis” and play an important role in the occurrence and development of liver cancer [7]. Several studies have suggested that intestinal microorganisms and their related metabolites interfere with immune checkpoints in various ways to regulate the immune responses in liver cancer [7]. Many previous studies have focused on the influence of intestinal flora in the development of liver cancer or focused on immunotherapy to improve the survival of patients with advanced liver cancer. Few studies have linked the possibility that the modulation of intestinal flora may enhance the effects of HCC immunotherapy. Therefore, this paper reviews the latest research progress on liver cancer immunotherapy; analyzes the change in the intestinal flora in patients liver cancer immunotherapy; and summarizes the probiotics, prebiotics, FMT, and rational use of antibiotics that can regulate the intestinal flora to enhance the effect of immunotherapy. Furthermore, it provides a basis for regulating intestinal flora and enhancing the efficacy and potential mechanism of immunotherapy for liver cancer.

## 2. Immunotherapy for Liver Cancer

Immunotherapy for liver cancer mainly involves killing tumor cells by activating the immune system, and it can be classified into two aspects. First, it can enhance the antitumor immune response by enhancing or activating the immune response mediated by the T cells. Second, it can selectively adjust and reshape the immune response in the tumor microenvironment according to the immune escape mechanism induced by the liver cancer, allowing T cells to recognize and kill tumor cells. Currently, immunotherapy for liver cancer includes ICIs, tumor vaccines, and chimeric antigen receptor T lymphocyte (CAR-T) therapy. ICIs are monoclonal antibodies that can prevent interactions between the checkpoint proteins in cancer and immune cells [8]. Notably, ICI therapy involves rebuilding the balance of the immune tolerance system and using the immune system to destroy tumor cells [9]. Furthermore, immune checkpoint inhibitors induce their antitumor effects by regulating the immune system [10]. ICIs include antibodies against programmed cell death protein-1 (PD-1), programmed cell death 1 ligand-1 (PD-L1), and cytotoxic T lymphocyte antigen-4 (CTLA-4). PD-1 is an immunosuppressed receptor expressed by CD8+ T cells and other immune cells during inflammation or infection [11]. The major ligands of PD-L1 are constitutively expressed on the surface of antigen-presenting cells and other tissues, including tumor cells [12]. Nivolumab is an anti-PD-1 antibody. When evaluated in the I/II non-randomized checkmate trial, 262 patients were included: 48 patients were in the dose-increasing stage, and 214 patients were in the dose-expanding stage. For nivolumab, the total remission disease control rate was 3%, and the patients receiving dose escalation rates were 64%, 3%, 15%, and 58%, respectively [13]. The trial also observed that the median remission duration of patients with initial sorafenib treatment was 17 months and that of patients who had previously received sorafenib treatment was 19 months. The 18-month overall survival rates were 57% and 44%, respectively [13]. In a clinical study, the median survival of liver cancer patients treated with nivolumab was found to be 16.4 months, and the objective response rate (ORR) of the nivolumab group (15 patients with complete response [CR]) was 14% [14]. In addition, pembrolizumab, an anti-PD-1 antibody, was evaluated in the phase II clinical study-224 trial, which enrolled 104 patients with advanced hepatocellular carcinoma who were intolerant to sorafenib or had progressed; 17 of 18 patients showed an immune response. Among them, 1 patient (1%) was in complete remission, and 17 (16%) were in partial remission. Simultaneously, 46 patients (44%) were stable, 34 (33%) progressed, and 6 (14%) could not be evaluated [15]. Furthermore, atezolizumab is an anti-PD-L1 antibody that is primarily evaluated in combination with bevacizumab, a vascular endothelial growth factor (VEGF) inhibitor. In a phase Ib trial, the ORR in patients with metastatic or unresectable hepatocellular carcinoma (HCC) was 34% [16]. Immunotherapy for liver cancer also includes tumor vaccines and CAR-T. Currently, the vaccines targeting liver cancer mainly include heterogeneous recombinant alpha-fetoprotein, dendritic cells (DCs), and oncolytic virus vaccines. Notably, a DNA vaccine targeting fibroblast activation proteins can change the tumor stroma, and the antitumor effect can be improved by cooperating with the tumor-antigen-specific DNA vaccine [17]. Moreover, DC tumor vaccines reinforce DCs, which can specifically recognize tumors induced or constructed in vitro into tumor patients and activate the immune response of the T lymphocytes to tumors. Furthermore, CAR-T couples the antigen-binding part of an antibody that can recognize a certain tumor antigen and the intracellular part of CD3 or FCR into a chimeric protein in vitro, and it transfects the patient’s T lymphocytes by gene transduction to express the chimeric antigen receptor. Common targets of adoptive cell therapy for liver cancer include alpha-fetoprotein, phosphatidylinositol proteoglycan-3, melanoma antigen gene-1/3, human telomerase reverse transcriptase, VEGF, epithelial cell adhesion molecule, and natural killer cells (NK) group 2 D ligand. Clinical studies on CAR-T cells for treating liver cancer with related targets have been reported. In solid tumors, overexpression of the basic leucine zipper transcription factor can block CAR-T cell depletion, enhance CAR-T cell proliferation and toxicity, and promote the formation of memory cells, thus providing long-term tumor treatment effects [18].

## 3. Effect of Intestinal Flora on the Immunotherapy of Liver Cancer

The intestinal flora comprises the largest microecosystem in the human body and contains more than 1014 types of microorganisms. It has observable effects on tumor immunotherapy and antitumor cell responses after chemotherapy [19,20]. Although the liver is not in direct contact with the microorganisms, it is closely connected to the intestine through the biliary tract, hepatic portal vein, and bile secretion. The intestine–liver axis plays an important role in the pathogenesis of HCC [21,22,23]. Ecological imbalance is the qualitative and quantitative changes in the intestinal microflora that may destroy the intestinal barrier and increase intestinal permeability. Ecological imbalance and intestinal leakage are associated throughout chronic liver disease. Furthermore, the ecological imbalance may help form a more permeable intestinal barrier. A leaky intestine makes it easier for bacterial metabolites and microbiota-related molecules connected with the imbalance to transfer to the liver and promote repeated inflammatory stimulation, fibrosis, and cirrhosis, eventually leading to liver cancer [24]. Interestingly, the intestinal flora plays an important role in the occurrence and development of liver cancer and can regulate innate and adaptive immune responses. Therefore, intestinal flora may play an important role in regulating the immunotherapeutic liver cancer. Previous studies have observed that before immunotherapy for liver cancer, there are significant differences in fecal bacteria between patients in objective remission (OR) and those in disease progression (PD). Among them, Prevost bacteria were enriched in patients with PD, whereas *Mucor, Chaetoceros*, and *Neisseria veyensis* were dominant in patients with OR. Furthermore, ursodeoxycholic acid and ursolic acid were abundant in the feces of patients with OR and are closely related to the abundance of *Clostridium pilosum*. Patients with better microbial characteristics had better survival times [25]. Moreover, Zheng [26] observed, through dynamic analysis of HCC patients treated with anti-PD-1 immunotherapy in the sixth week, that there was a significant difference in the β diversity of the intestinal microorganisms between the immune responders and the non-responders, and the fecal samples of patients who responded to immunotherapy showed higher taxonomic richness and gene counts. With the progress of immunotherapy, from the phylum level, the microbial composition of the responders remained relatively stable and was enriched with myxobacteria and rumen cocci. However, the number of *Proteus* spp. increased significantly in the third week, and they became the dominant bacteria in the twelfth week. According to the findings, dynamic changes in the intestinal microbial diversity and composition during anti-PD-1 immunotherapy for HCC may affect drug efficacy and disease prognosis [26]. Therefore, manipulating the intestinal microflora may have a positive impact on the effect of immunotherapy on liver cancer.

## 4. Drug Resistance Characteristics of Immunotherapy

During the process of immunotherapy, some patients experience a remarkable curative effect. However, some still develop the disease after a period of remission, suggesting the existence of immune resistance. Thus, drug resistance mechanisms in immunotherapy can be classified into primary drug resistance, adaptive drug resistance, and acquired drug resistance. Primary drug resistance means that the patient does not respond to immunotherapy, and T cells cannot recognize the tumor, leading to a lack of tumor antigens [27]. There are three main aspects of drug resistance mechanisms. The first is that the immunogenicity of the tumor itself is poor, escaping immune system surveillance, and tumor neoantigen presentation is inhibited, resulting in a tumor microenvironment conducive to tumor growth and the inhibition of immune infiltration. The second is an abnormal signaling pathway, including PTEN deletion, Wnt/β-catenin abnormal activation, and IFN pathway suppression. The third is that the intestinal flora are associated with tumor immunotherapy resistance. Acquired drug resistance refers to the initial efficacy of immunotherapy. However, the tumor recurs and progresses after a period, which may result from the clonal selection of heterogeneous groups and drug-resistant cells before treatment begins. Possible mechanisms of drug resistance include the loss of T-cell function, a mutation of the β2-microglobulin gene, a change in the tumor target antigen, a mutation of the Apelin receptor gene, and an imbalance of intestinal flora. This indicates that an imbalance in the intestinal flora may affect the effectiveness of immunotherapy and lead to drug resistance [28]. Arielle et al. [29] showed that antibiotic-induced flora imbalance can reduce the activation of the immune system by ICIs, leading to drug resistance and poor curative effects. Moreover, Vetizou [30] has shown that *Bacteroides fragilis* and *Bacteroides thetaiotaomicron* can regulate specific T-cell responses. In addition, oral *Bifidobacterium* exerts a positive effect by increasing the response to immunotherapy. Compared with fecal microflora transplantation (FMT) in unresponsive patients, responder stool transfer helps restore the anticancer effect blocked by immunotherapy with PD-1 and CTLA-4 [31].

## 5. Effect of Regulating Intestinal Flora on Immunotherapy

Regulating the intestinal flora by transplantation of probiotics, prebiotics, and fecal microflora or the rational use of antibiotics may reduce drug resistance and the related complications of immunotherapy and enhance the antitumor effect of immunotherapy drugs. Recently, an increasing number of studies have shown that intestinal flora plays an important role in tumor occurrence and regulation of tumor treatment, especially in improving the effectiveness of immunotherapy drugs [32,33,34].

### 5.1. Effect of Probiotic Therapy on Immunotherapy

Probiotics are active microorganisms that colonize the human body and change the composition of the flora in certain parts of the host in a manner that is beneficial for the host. Probiotics can regulate the immune function of the host mucosa and system or optimize the epithelial barrier function by regulating the balance of the intestinal flora. They can limit the damage that pathogenic bacteria can cause by reducing pathogen binding sites, promoting cellular protective responses, increasing mucin secretion, and reducing inflammation in the intestinal tract [35,36]. Common probiotics include yeast, *Clostridium butyricum*, *Lactobacillus*, *Bifidobacterium*, and *Actinomycetes*. Studies have shown that during cancer treatment, probiotics can restore the balance of intestinal microorganisms and reduce the occurrence of immunotherapy-related complications [37]. In addition, Sivan et al. observed that the oral administration of *Bifidobacterium* and *Enterobacter* in mice promoted the maturation of DCs and increased the number of CD8+ T cells, which restored the antitumor effect of the PD-L1 blockade [38]. *Bifidobacterium* can also mediate innate immunity, secrete the metabolite hippurate, and inhibit the expression of PD-1, thus activating NK cells and eliminating tumors through perforin and IFNγ [39]. Moreover, *Lactobacillus rhamnosus* had a synergistic effect with the immune checkpoint blockade treatment. Oral *L. rhamnosus* can enhance the antitumor activity of anti-PD-1 immunotherapy by increasing the tumor infiltration of DCs and T cells. The antitumor effect in the group receiving probiotic treatment was significantly improved compared with the group without probiotic treatment [40,41,42]. Similarly, Gao et al. confirmed these findings and observed that probiotic *L. rhamnosus*-M9 could improve the efficacy and responsiveness of immunotherapy based on anti-PD-1 [40]. Interestingly, *Lactobacillus paracasei* sh2020 stimulation triggered the upregulation of CXCL10 expression in tumors and increased CD8+ T cells to enhance the antitumor activity of anti-PD-1 immunotherapy and reduce the tumor load in mice [43].Probiotics can also enhance the effect of immunotherapy by interacting with antioxidants to prevent and treat certain cancers [44,45]. Diet-derived polyphenols are bioactive compounds and are a common type of antioxidant [46]. Dietary supplementation of polyphenols can significantly induce the production of probiotics such as Bifidobacterium and Lactobacillus in the host intestine and inhibit the production of harmful bacteria such as *Clostridium* and *Escherichia coli* [47]. Polyphenols and their active metabolites can enhance the production of short-chain fatty acids (SCFAs) and branch-chain amino acids and reduce gastrointestinal inflammation by down-regulating pro-inflammatory cytokines. Therefore, it can be used to prevent and treat a variety of diseases, including gastrointestinal inflammatory diseases, gastric cancer, and colon cancer [47]. The chemoprophylaxis of polyphenols can prevent cancer, reduce tumorigenesis, and affect pathways related to cell proliferation by enhancing the cytotoxic effects on cancer cells while reducing the toxicity to normal surrounding tissues [44,47]. Probiotics have a synergistic effect with antioxidants. Caponio GR et al. found that grape dregs (GP) contain a large number of phenolic compounds, which can produce bioactive compounds with antioxidant, antibacterial, and anti-inflammatory activities. In vitro simulated gastrointestinal digestion of GP extract, *L. monocytogenes* decreased but *E. coli* and *B. megaterium* grew only when inoculated with probiotic *L. plantarum*. When probiotics (*L. plantarum*) were combined with GP antioxidants, decreases in *E. coli*, *B. megaterium*, and *L. monocytogenes* were observed [45].

### 5.2. Influence of Prebiotic Therapy on Immunotherapy

Prebiotics are dietary supplements. They generally refer to organic substances that are not digested and absorbed by the host, but that can selectively promote and stimulate the growth and reproduction of one or a few colonies (such as *Bifidobacterium*) and other beneficial bacteria, thus improving the health of the host. They mainly include functional oligosaccharides (such as fructooligosaccharides, galactooligosaccharides, xylooligosaccharides, isomaltooligosaccharides, soybean oligosaccharides, and inulin), polysaccharides, extracts of natural plants (such as vegetables, Chinese herbal medicines, and wild plants), protein hydrolysates, and polyols [48]. In several mouse tumor models, a gel made of inulin can regulate the intestinal microflora, induce systemic memory T-cell responses, and amplify the antitumor activity of checkpoint inhibitors against α-PD-1. Furthermore, oral inulin gels can promote the relative abundance of key symbiotic microorganisms and their short-chain fatty acid metabolites [49]. Moreover, resistant starches promoted the growth of bacteria involved in butyrate production. Notably, Taper et al. incorporated fructooligosaccharides or inulin into the basic diet of mice with liver cancer, significantly improving the curative effect of cytotoxic drugs [50]. Additionally, inosine can activate type 1 helper (Th1) cells through the adenosine 2A receptor, stimulated by DCs, and enhance the therapeutic effect on tumors in the presence of ICIs [51].

### 5.3. Effect of FMT Treatment on Immunotherapy

FMT involves mixing donor feces with normal saline and transplanting them into patients to change their intestinal microflora. This approach has been used to treat various gastrointestinal diseases [52,53,54]. The composition of the intestinal microflora affects the response to anti-PD-L1 therapy. Transplantation of fecal microflora from ICI-responsive patients (FMT) into sterile or antibiotic-treated mice can improve tumor control and the response to ICIs [31,55]. Following FMT, the ICI-unresponsive patients can recover their anti-PD-1 treatment response through oral supplementation with viscous Ackermann flora [31]. Furthermore, the ICI non-responders can restore the effect of the PD-1 blockade in an interleukin-12-dependent manner by increasing the recruitment of CCR9+CXCR3+CD4+T lymphocytes to the tumor beds of mice [39].

### 5.4. Effect of Antibiotic Therapy on Immunotherapy

Antibiotics have several adverse effects on the intestinal microflora, including decreased species diversity, changes in metabolic activity, and the selection of antibiotic-resistant microorganisms [56]. Previous studies have mainly evaluated the effect of antibiotics on ICI efficiency and indicated that the use of antibiotics might weaken the benefits of immunotherapy and reduce survival time [31,57,58]. For instance, Ahmed [59] reported that patients who received antibiotic treatment within two weeks before and after ICI treatment took longer to respond to the treatment and that their response rate and survival time were lower than those who did not. Furthermore, Routy et al. showed that the survival time of cancer patients treated with an anti-PD-1/PDL-1 antibody was significantly shortened after oral treatment with antibiotics [31]. Moreover, Xu [60] reported that after mice were treated with the PD-1 antibody, the injection of antibiotics counteracted the inhibitory effect of the PD-1 antibody on tumor growth compared with the control group (mice that were treated with sterile drinking water). Finally, a meta-analysis by Lurienne et al. showed that using antibiotics before or during ICI treatment reduced the median survival time by >6 months [61].

## 6. Potential Mechanism of Intestinal Microflora Affecting Immunotherapy

There are three aspects by which the intestinal microflora affects cancer immunotherapy. First, the intestinal microflora directly stimulates the antitumor T-cell response. Second, a common antigen mimic is formed through the bacterial surface antigen and tumor cell antigen, which triggers a cross-antitumor T-cell reaction. Third, metabolites of the intestinal microflora regulate antitumor immunotherapy.

### 6.1. Direct Stimulation of Antitumor T-Cell Response

Some gut microbiota can trigger specific antitumor T-cell responses. Figure 1 illustrates the following: first, *Helicobacter hepatogenes* stimulate RORγtFOXP3 to regulate T cells that selectively inhibit pro-inflammatory T-helper 17 (Th17) cells and their functions that rely on transcription factor c-MAF, enhancing interleukin (IL)-10, IL-23, and T-cell expression; and second, *Akkermansia muciniphila* modulates PD-1 checkpoints, inducing B cells to produce immunoglobulin G1 antibodies and antigen-specific T cells acting on mouse tumor cells [62,63]. Fragile bacilli and their components directly or indirectly stimulate phagocytes to develop an immune response and produce interferon, which can enhance the efficacy of CTLA-4 treatment [64]. *Bacillus flattiformis* can directly enhance the number of CD8+ T cells in tumor cells after cyclophosphamide treatment. Figure 1D shows that *Bifidobacterium* enhances the expression of T-cell markers to produce IFNγ, which acts on tumor cells [65]. *Bifidobacterium* in rodents directly or indirectly stimulates macrophages. Macrophages activates the stimulator of interferon genes and IFN, which produces IFN1, IFNγ, tumor necrosis factor (TNF)-α, and IL-2. It enhances CD8+ T-cell expression, upregulates PD-L1 expression in the mouse tumor cell, and increases antibody CD47 expression, which sensitizes mouse tumor cells to anti-CD47 immunotherapy [66,67]. In addition, bacterial flagellin can directly stimulate the expression of Toll-like receptors 5 and 4 on the surface of dendritic cells and macrophages. DCs act on T cells, which produce IFNα, IL-8, and IL-6; enhance the host immune response; and inhibit tumor cell growth. This is evidenced by flagellar proteins secreted by *Enterococcus gallinaurus* and *Salmonella typhimurium* [68,69].

### 6.2. Initiate Cross-Antitumor T-Cell Reaction by Forming a Common Antigen Mimic with Tumor Cells

The intestinal microbial surface antigens and tumor cell surface antigens are simulated by molecules to form common antigens to trigger cross-reactive T cells. As shown in Figure 2, the antigen on the surface of the commensal bacterium *Bifidobacterium breve,* called SVYRYYGL (SVY), undergoes T-cell cross-reactivity with the mouse tumor model neoantigen SIYRYYGL (SIY). *Bifidobacterium* promotes the expression of SVY-responsive T cells, acts on the SIY on the surface of tumor cells, enhances CD8+ T-cell expression, inhibits and kills tumor cells, reduces tumor growth, and prolongs survival. The tail long tape measure protein (TMP) of the *Enterococcus hirae* phage and the tumor cell surface antigen have a common antigen TMP, which triggers a crossover antitumor T-cell response, enhances CD8^+^ T-cell expression, and increases the therapeutic effect of PD-1 blockers [70,71]. The long-term survival of patients with pancreatic cancer is related to developing new antigens with high immunogenicity and cross-reactivity with microbial epitopes [64]. The FAP2 protein of *Fusobacterium nucleatum* interacts with the inhibitory T-cell receptor TIGIT. TIGIT is an inhibitory receptor found on all human NK cells and various T cells, which can directly inhibit the expression of NK and T-cell markers, thereby inhibiting the antitumor cell immune response [72].

### 6.3. Metabolites Regulate Antitumor Immunotherapy

Small-molecule metabolites directly synthesized or indirectly transformed by the intestinal microbiota can spread in the original location in the intestine and affect local and systemic antitumor immune responses. They can improve the efficacy of ICIs and play an immunomodulatory role. Common immunomodulatory gut microbiota metabolites include inosine, bile acids, and SCFAs. Inosine is a purine metabolite of *Bifidobacterium* that can reshape the tumor microenvironment and improve the response to ICI therapy. As shown in Figure 3, inosine mainly affects the efficacy of ICIs through the following mechanisms: first, inosine enhances the ability of tumor cells to present tumor antigens, promotes the activity of tumor-specific T cells to recognize and kill tumor cells, and achieves antitumor effects; and second, adenosine regulates the differentiation and accumulation of Th1 cells by stimulating DCs and adenosine receptors acting on the surface of T cells, producing IFNγ and IL-12 and improving the anti-PD-1 therapeutic response [73,74]. Inosine can also be used as an energy substrate for CD8+ T cells, providing them with energy and improving the antitumor effect of T cells [75]. Bile acids may play an immunosuppressive role in antitumor immunotherapy. As an important component of the intestine–liver axis, primary bile acids produced in the liver are secreted into the small intestine and metabolized into secondary bile acids by intestinal microorganisms to promote the digestion and absorption of lipids and certain vitamins [76]. Studies have found that the type of secondary bile acids depends on the composition of the gut microbiota, and some secondary bile acids are closely associated with liver lesions [77,78]. As shown in Figure 4, primary bile acids can be converted into secondary bile acids and influence immunotherapy by altering gut microorganisms, thereby promoting the number of hepatic CXCR6 + NKT cells, weakening the selective tumor suppression effect of the liver, and possibly inhibiting and killing liver cancer cells [79]. Studies have shown that the bile acid metabolites 3-oxocholic acid and isochoric acid alter the biochemical modification of intestinal microorganisms and can inhibit the differentiation of Th17 cells [80]. When the secondary bile acid 3β-hydroxydeoxycholic acid acts on DCs, it weakens immune stimulation, induces the expression of Foxp3, increases the number of regulatory T cells, and promotes immune escape [81]. Another important metabolite of gut microbes is SCFAs. The byproducts of dietary fiber bacterial fermentation of SCFAs in the colon include acetate, propionate, and butyrate, among others [82]. SCFAs provide energy to immune cells, improve antitumor immune responses, and affect regulatory T cells, effector T cells, and γδ T-cell expression [83,84,85,86]. As shown in Figure 5, butyrate can directly act on DCs and CD8+ T cells to produce IL-12 and enhance the antitumor cytotoxicity of specific T cells. Valerate co-butyrate acts on antigen-specific cytotoxic T lymphocytes and CAR-T cells, significantly enhancing the content of CD25, IFNγ, and TNF-α factors and promoting antitumor effects. In two separate studies of anti-PD-1 therapy in cancer patients, a high expression of SCFAs produced a positive immune response [87,88].

Inosine mainly affects the efficacy of ICIs through the following mechanisms: first, inosine enhances the ability of tumor cells to present tumor antigens, promotes tumor-specific T cells to recognize and kill tumor cells, and achieves an antitumor effect; second, adenosine stimulates DCs and A2R acting on the surface of T cells to produce IFNγ and IL-12 to regulate the differentiation and accumulation of Th1 cells and improve anti-PD-1 therapy. Inosine can be used as an energy substrate for CD8+ T cells, providing energy for them and improving the antitumor effect of T cells.

After the gut microorganism converts primary bile acids into secondary bile acids, it will return to the liver through the enterohepatic circulation, thereby promoting the expression of hepatic CXCR6 + NKT cells, weakening the selective tumor suppressor effect of the liver, and potentially inhibiting and killing liver cancer cells. The biochemically modified bile acid metabolites 3-oxocaryocholic acid and isocholic acid of intestinal microorganisms can inhibit the differentiation of Th17 cells. When the secondary bile acid 3β-hydroxydeoxycholic acid acts on dendritic cells, it weakens immune stimulation, induces the expression of Foxp3, upregulates the number of Treg, and promotes immune escape.

The products of dietary fiber bacterial fermentation of short-chain fatty acids (SCFAs) are acetate, propionate, and butyrate. Furthermore, SCFAs can affect Tregs, enhance the expression of effector T cells and γδ T cells, and improve antitumor immune response. Butyrate can directly act on DCs and CD8+ T cells to produce IL-12 and enhance the antitumor cytotoxicity of specific T cells. Valerate co-butyrate acts on antigen-specific CTLs and CART cells to enhance the expression of CD25, IFNγ, and TNF-α, thus promoting antitumor effects.

## 7. Conclusions

Immunotherapy is an important research direction in treating liver cancer because it can prolong the survival time of patients with liver cancer to a certain extent. However, there are individual differences, and patients with liver cancer exhibit poor immune responses. Furthermore, intestinal microorganisms and their derived metabolites can regulate local and systemic antitumor immune responses, thus improving the immune response and immunotherapeutic effect of patients with liver cancer. With the development of animal experiments and clinical studies, it has been observed that probiotics, prebiotic supplements, and FMT are likely to be strategies to improve the efficacy of immunotherapy for liver cancer. Simultaneously, it has been observed that a reasonable selection of antibiotics and bacterial genetic engineering might provide a theoretical basis for the intestinal flora and its metabolites to enhance the immunotherapeutic effect liver cancer. A clinical trial of the intestinal flora to improve drug resistance in liver cancer immunotherapy is recruiting patients and is expected to clarify how to use intestinal flora to enhance the effect of immunotherapy. This will help understand the role of intestinal flora in the antitumor immunotherapy of liver cancer, provide a new individualized treatment model, and thus improve the clinical efficacy of immunotherapy for liver cancer.

## Figures and Tables

**Figure 1 ijms-24-11387-f001:**
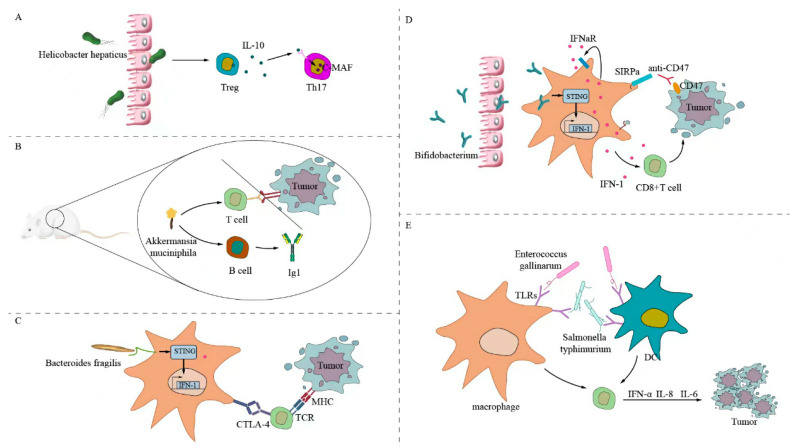
Gut microbiota directly stimulates the antitumor T-cell response. (**A**) *Helicobacter hepaticus* directly stimulates RORγtFOXP3 to regulate T cells to selectively inhibit TH17 cells. (**B**) *Akkermansia muciniphila* stimulates antigen-specific T cells and B cells to produce IgG1 antibodies, which can regulate PD-1 checkpoints and enhance antitumor effects. (**C**) Fragile bacilli and components directly or indirectly stimulate macrophages, activate STING to produce interferon, and can enhance the efficacy of CTLA-4 treatment. (**D**) Bifidobacterium can directly act on macrophages; activate STING and IFN; produce IFN1, IFNγ, TNF-α, and IL-2; enhance CD8+ T cell expression; upregulate PD-L1 expression on the tumor cell surface; enhance antibody CD47 expression; and inhibit the growth of mouse tumor cells. (**E**) Bacterial flagellar proteins can directly stimulate the expression of TLR5 and TLR4 on the surface of DCs and macrophages; act on T cells; produce IFNα, IL-8, and IL-6; enhance the host immune response; and inhibit tumor cell growth.

**Figure 2 ijms-24-11387-f002:**
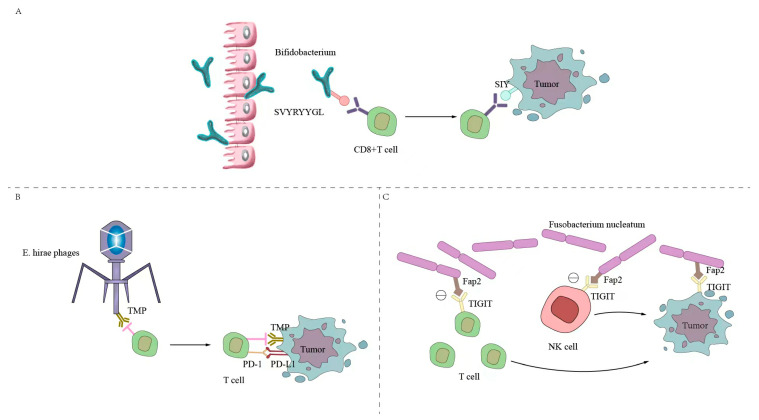
Intestinal microbiota initiates cross-antitumor T-cell responses by forming a shared antigenic mimic with tumor cells. (**A**) The SVYRYYGL (SVY) on the surface of *Bifidobacterium breve* cross-reacts with the mouse tumor model neoantigen SIYRYYGL (SIY). *Bifidobacterium* promotes the expression of SVY-responsive T cells, acts on the SIY on the surface of tumor cells, enhances the expression of CD8+ T cells, and inhibits the killing of tumor cells. (**B**) The *Enterococcus hirae* bacteriophage has a common antigen, TMP, with the tumor cell surface antigen, which triggers a crossover antitumor T-cell response, enhances CD8+ T-cell expression, and enhances the efficacy of PD-1 blockers. (**C**) The FAP2 protein of *Fusobacterium nucleatum* interacts with the inhibitory T-cell receptor TIGIT, which can directly inhibit the expression of NK cells and T cells, thereby inhibiting the immune response of antitumor cells.

**Figure 3 ijms-24-11387-f003:**
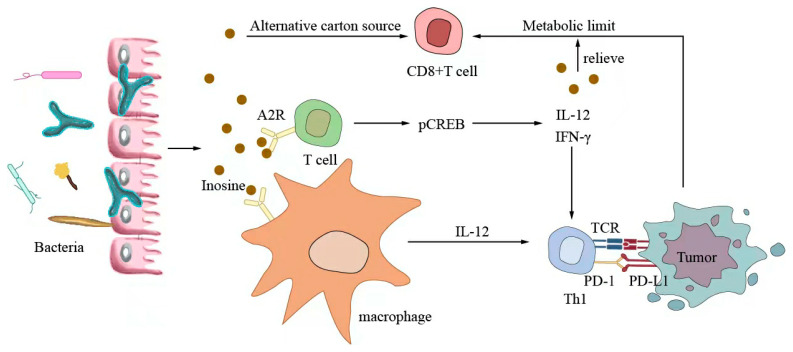
Inosine-modulated antitumor immunotherapy.

**Figure 4 ijms-24-11387-f004:**
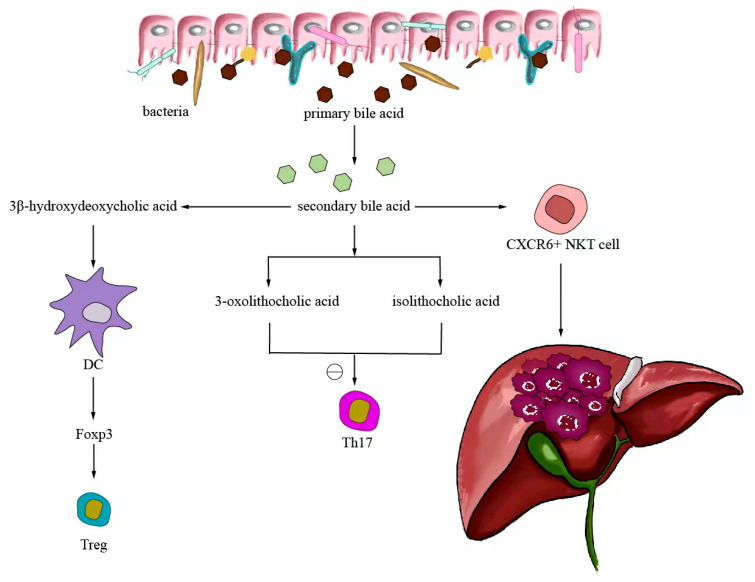
Bile acid-regulated antitumor immunotherapy.

**Figure 5 ijms-24-11387-f005:**
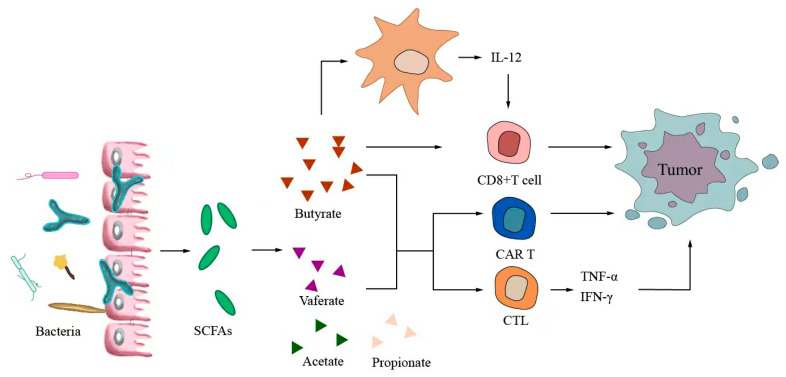
Short-chain fatty acids modulate antitumor immunotherapy.

## Data Availability

This is a review article and does not contain any research data.

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
