# Peer review of "Potential Effects of Regulating Intestinal Flora on Immunotherapy for Liver Cancer"

_ijms, 2023, doi:10.3390/ijms241411387_

Round 1

Reviewer 1 Report

In the review entitled “Research progress of intestinal flora regulating immunotherapy for liver cancer” the authors focused on the relationship between intestinal flora and immunotherapy for liver cancer and further clarifies the potential relationship between intestinal flora and immunotherapy for liver cancer. This is an interesting topic, and it is an area that really needs our attention. Overall, the work is well done, but there are still some areas of the article that need to be revised.

1. Keywords:
-Please write in alphabetic order.

2. Introduction:

- The concept of intestinal flora and related metabolites in the regulation of metabolic processes in chronic hepatitis and liver cancer through the "intestinal-liver axis" lacks bibliographic references. I suggest enriching the scientific relevance.

-Overall, the introduction lacks an emphasis on the novelty of this review. The authors should compare the scientific relevance with this study in order to demonstrate how this review is different from those that have already been published and improve to better describe the innovativeness.

3. Please include a section after the introduction, defining, the method to collect literature, what was the time domain, which data sources were considered. 

4. In order to keep attention to the recent challenges to use diet and bioactive compounds in several diseases, I suggest adding a paragraph about the synergic use of antioxidants and probiotics for the prevention and treatment of certain cancer. Perhaps these articles are helpful to be cited and discussed:

-Caponio, G. R., Lippolis, T., Tutino, V., Gigante, I., De Nunzio, V., Milella, R. A., ... & Notarnicola, M. (2022). Nutraceuticals: Focus on anti-inflammatory, anti-cancer, antioxidant properties in gastrointestinal tract. Antioxidants, 11(7), 1274.

-Lippolis, T., Cofano, M., Caponio, G. R., De Nunzio, V., & Notarnicola, M. (2023). Bioaccessibility and bioavailability of diet polyphenols and their modulation of gut microbiota. International Journal of Molecular Sciences, 24(4), 3813.

-Thyagarajan, A., & Sahu, R. P. (2018). Potential contributions of antioxidants to cancer therapy: immunomodulation and radiosensitization. Integrative cancer therapies, 17(2), 210-216.

-Caponio, G. R., Minervini, F., Tamma, G., Gambacorta, G., & De Angelis, M. (2023). Promising Application of Grape Pomace and Its Agri-Food Valorization: Source of Bioactive Molecules with Beneficial Effects. Sustainability, 15(11), 9075.

5. The language expression needs to be improved throughout the manuscript. 

Minor editing of English language required

Author Response

Dear reviewer,

First of all, I would like to express my sincere gratitude to you and the reviewers for providing valuable feedback on my paper. As a result, I have made revisions to the manuscript to better address your and the reviewers' comments. Below are our responses to each of your suggestions:

In the review entitled “Research progress of intestinal flora regulating immunotherapy for liver cancer” the authors focused on the relationship between intestinal flora and immunotherapy for liver cancer and further clarifies the potential relationship between intestinal flora and immunotherapy for liver cancer. This is an interesting topic, and it is an area that really needs our attention. Overall, the work is well done, but there are still some areas of the article that need to be revised.

  1. Keywords:
    -Please write in alphabetic order.

Thank you for your comment and suggestions.We have carefully considered your feedback and revised the keywords to write them alphabetically according to your requirements.

  1. Introduction:

- The concept of intestinal flora and related metabolites in the regulation of metabolic processes in chronic hepatitis and liver cancer through the "intestinal-liver axis" lacks bibliographic references. I suggest enriching the scientific relevance.

-Overall, the introduction lacks an emphasis on the novelty of this review. The authors should compare the scientific relevance with this study in order to demonstrate how this review is different from those that have already been published and improve to better describe the innovativeness.

Following your revised requirements, I added references to the concept that intestinal flora and related metabolites regulate metabolic processes in chronic hepatitis and liver cancer through the "enterohepatic axis". Compared to this study, many previous studies have focused on the effects of the gut microbiota on the development and progression of hepatocellular carcinoma, or immunotherapy to improve survival in patients with advanced hepatocellular carcinoma. Few studies have linked modulating the gut microbiota to possibly enhancing the efficacy of immunotherapy for hepatocellular carcinoma. This article highlights the effect that regulating intestinal flora may affect the effect of immunotherapy for liver cancer, and provides relevant theoretical basis for clinical practice. 

  1. Please include a section after the introduction, defining, the method to collect literature, what was the time domain, which data sources were considered. 

We searched the reference literature for nearly 10 years through PubMed and found that the intestinal flora plays a major role in the occurrence and development of liver cancer, and the intestinal flora of patients with intermediate and advanced liver cancer has changed after immunotherapy, and the intestinal flora can change the effect of cancer immunotherapy. Therefore, we explored whether intervention in the intestinal flora can have an impact on liver cancer immunotherapy. All references are imported into the article via endnote. All references are relevant to the literature.

  1. In order to keep attention to the recent challenges to use diet and bioactive compounds in several diseases, I suggest adding a paragraph about the synergic use of antioxidants and probiotics for the prevention and treatment of certain cancer. Perhaps these articles are helpful to be cited and discussed:

-Caponio, G. R., Lippolis, T., Tutino, V., Gigante, I., De Nunzio, V., Milella, R. A., ... & Notarnicola, M. (2022). Nutraceuticals: Focus on anti-inflammatory, anti-cancer, antioxidant properties in gastrointestinal tract. Antioxidants, 11(7), 1274.

-Lippolis, T., Cofano, M., Caponio, G. R., De Nunzio, V., & Notarnicola, M. (2023). Bioaccessibility and bioavailability of diet polyphenols and their modulation of gut microbiota. International Journal of Molecular Sciences, 24(4), 3813.

-Thyagarajan, A., & Sahu, R. P. (2018). Potential contributions of antioxidants to cancer therapy: immunomodulation and radiosensitization. Integrative cancer therapies, 17(2), 210-216.

-Caponio, G. R., Minervini, F., Tamma, G., Gambacorta, G., & De Angelis, M. (2023). Promising Application of Grape Pomace and Its Agri-Food Valorization: Source of Bioactive Molecules with Beneficial Effects. Sustainability, 15(11), 9075.

Thank you very much for your advice, by reviewing the literature, we found that antioxidants in the diet have a synergistic effect with probiotics to enhance the effect of immunotherapy for certain cancers. At the end of the first subsection of the fifth part of our article, a paragraph on the synergy between probiotics and antioxidants has been added to enrich the structure of the article. At the same time, it also shows that the daily diet pattern can adjust probiotics, enhance the effect of immunotherapy, make up for my shortcomings, and make the article more readable and scientific. 

  1. The language expression needs to be improved throughout the manuscript. 

We have carefully reviewed the manuscript and have made revisions to ensure that the language is clear, concise, and appropriate for an academic publication. All the modifications have changed the black font to the red font and marked it, so please check it in your busy schedule.

Thank you very much for your valuable suggestions and feedback. I hope that my revisions meet your requirements. If you have any further suggestions or need further clarification, please do not hesitate to let me know. Thank you again for your attention and support.

Best regards,

Xiangdong Yan

Reviewer 2 Report

Dear Authors, 

I would like to congratulate you for your submission of the manuscript "Research progress of intestinal flora regulating immunotherapy for liver cancer". 

I find the manuscript overall well written, however there are a few minor corrections that might improve the readability of your work for the scientific community. I hope you find the suggestions useful in the review process:

1. Please reconsider the title. I would refrain from using the words "research progress". 

2. Your paragraphs are as long as the sections. Please break the long paragraphs into separate shorter ones, based on the main idea discussed in the sentences. When the idea changes, please insert a new paragraph within the section. 

Additionally, in case the figures are not personal contribution, please make sure you are allowed to adapt them from existent publications. 

All in all, the manuscript is, in my opinion, a good review, with more than 50% of the references used being published within the past 5 years (after 2018). 

Best of luck and best regards, 

One of your Reviewers

English language is fine. 

Paragraph length should be reduces. One paragraph should only focus on one aspect. 

Author Response

尊敬的审稿人,

首先,我要衷心感谢您和审稿人对我的论文提供宝贵的反馈。因此,我对手稿进行了修改,以更好地解决您和审稿人的意见。以下是我们对您的每个建议的回应:

对作者的意见和建议

亲爱的作者,

祝贺您投稿《肝癌肠道菌群调控免疫治疗研究进展》。

我发现手稿总体上写得很好,但是有一些小的更正可能会提高你的工作对科学界的可读性。我希望您在审查过程中发现这些建议有用:

  1. 请重新考虑标题。我不想使用“研究进展”一词。

感谢您的意见和建议。我们仔细考虑了您的反馈,并修改了标题以更好地突出本文的主题,将文章标题更改为“调节肠道菌群对肝癌免疫治疗的潜在影响”。此外,我们在引言末尾添加了本文的创新,以更好地引导读者阅读稿件。

  1. 您的段落与部分一样长。请根据句子中讨论的主要思想,将长段落分成单独的较短段落。当想法发生变化时,请在该部分中插入一个新段落。

此外,如果这些数字不是个人贡献,请确保您被允许从现有出版物中改编它们。

我们仔细审查了手稿并进行修改,以确保语言清晰、简洁且适合学术出版物。根据您的要求,我们在文章中将长句转换为短句,并添加了连词和代词以使文章更具可读性,并修改了一些语言。文章的具体内容用红色字体标注。

非常感谢您的宝贵建议和反馈。我希望我的修订符合您的要求。如果您有任何进一步的建议或需要进一步澄清,请随时告诉我。再次感谢您的关注和支持。

此致敬意

闫向东
